# Chloride Intracellular Channel Proteins (CLICs) and Malignant Tumor Progression: A Focus on the Preventive Role of CLIC2 in Invasion and Metastasis

**DOI:** 10.3390/cancers14194890

**Published:** 2022-10-06

**Authors:** Saya Ozaki, Kanta Mikami, Takeharu Kunieda, Junya Tanaka

**Affiliations:** 1Department of Neurosurgery, Graduate School of Medicine, Ehime University, Toon 791-0295, Japan; 2Department of Neurosurgery, National Cerebral and Cardiovascular Center Hospital, Suita 564-8565, Japan; 3Department of Molecular and Cellular Physiology, Graduate School of Medicine, Ehime University, Toon 791-0295, Japan

**Keywords:** metastasis, invasion, MMP, MT1-MMP, CLIC4, glioma, benign tumor, tight junction

## Abstract

**Simple Summary:**

Although chloride intracellular channel proteins (CLICs) have been identified as ion channel proteins, their true functions are still elusive. Recent in silico analyses show that CLICs may be prognostic markers in cancer. This review focuses on CLIC2 that plays preventive roles in malignant cell invasion and metastasis. CLIC2 is secreted extracellularly and binds to matrix metalloproteinase 14 (MMP14), while inhibiting its activity. As a result, CLIC2 may contribute to the development/maintenance of junctions between blood vessel endothelial cells and the inhibition of invasion and metastasis of tumor cells. CLIC2 may be a novel therapeutic target for malignancies.

**Abstract:**

CLICs are the dimorphic protein present in both soluble and membrane fractions. As an integral membrane protein, CLICs potentially possess ion channel activity. However, it is not fully clarified what kinds of roles CLICs play in physiological and pathological conditions. In vertebrates, CLICs are classified into six classes: CLIC1, 2, 3, 4, 5, and 6. Recently, in silico analyses have revealed that the expression level of CLICs may have prognostic significance in cancer. In this review, we focus on CLIC2, which has received less attention than other CLICs, and discuss its role in the metastasis and invasion of malignant tumor cells. CLIC2 is expressed at higher levels in benign tumors than in malignant ones, most likely preventing tumor cell invasion into surrounding tissues. CLIC2 is also expressed in the vascular endothelial cells of normal tissues and maintains their intercellular adhesive junctions, presumably suppressing the hematogenous metastasis of malignant tumor cells. Surprisingly, CLIC2 is localized in secretory granules and secreted into the extracellular milieu. Secreted CLIC2 binds to MMP14 and inhibits its activity, leading to suppressed MMP2 activity. CLIC4, on the other hand, promotes MMP14 activity. These findings challenge the assumption that CLICs are ion channels, implying that they could be potential new targets for the treatment of malignant tumors.

## 1. Introduction

### 1.1. CLIC Family

Although chloride intracellular channel proteins (CLICs) have been identified as chloride ion channels, it remains unclear whether they indeed function as ion channels [1,2,3]. In vertebrates, there are six CLIC family members (CLIC 1–6) with well-conserved molecular structures (Table 1). Among the six CLICs, CLIC1 and CLIC4 are the most widely studied with respect to their localization, functions, and expression [1,3]. CLICs are localized in soluble fractions in the cytosol and nuclei rather than in membrane fractions [3,4,5]. CLICs are widely expressed in metazoans and have well-conserved molecular structures. Therefore, CLICs are assumed to play an important role in physiological processes. Many roles of CLICs other than acting as ion channels have been proposed [6,7,8,9], although a recent study reported that CLICs activate NOD-, LRR-, and pyrin domain-containing protein 3 (NLRP3) inflammasome probably by increasing the efflux of chloride ions [10].

CLICs are relatively small globular proteins with a molecular mass of approximately 30 kDa. Because CLICs possess a molecular structure resembling glutathione-S-transferase (GST), they have been investigated for enzymatic activity, although most researchers do not support this view [3,34]. In addition to the formation of chloride ion channels [10], CLICs are thought to play various roles, including ryanodine receptor (RyR) modulation [18,35,36,37], plasma membrane remodeling [38], intracellular trafficking [29,39], intracellular tubule formation [7], actin cytoskeleton reorganization [9], inflammasome activation [10], and TGFβ-mediated signal modification [40,41]. CLICs have been found in various intracellular locations, including the cytoplasm, mitochondria, endosome, nuclei, endoplasmic reticulum, and secretory granules [3,4,5,42,43]. CLICs are mainly found in soluble fractions, but they also occur in membrane fractions [1,3,34]. It is necessary to elucidate how these two localizations are regulated [44]. Recombinant CLICs are well known for their high solubility in aqueous buffers [1]. CLIC2 is secreted extracellularly in significant amounts, and its role in the extracellular fluid has been postulated, as described below; CLIC4 is also secreted in small amounts [4]. Regarding the distribution, functions, and structures of CLICs expressed in vertebrates, see Table 1 and well summarized reviews [3,42,45,46,47].

### 1.2. Known Structures of CLIC2

CLICs exist as both soluble globular proteins and integral membrane proteins that potentially possess ion channel activity. Because of the presence of reactive cystein residues, pH and redox conditions in most instances affect the transition between these two states [1]. Highly soluble recombinant CLIC proteins possess a putative transmembrane domain. When recombinant CLIC proteins are added to artificial synthetic lipid bilayers, they are integrated into the bilayer and reproducible ion channel activity is detected by electrophysiological measurements. CLIC2 is also demonstrated to form ion channels in lipid bilayers in a pH-dependent manner; a marked ion channel activity of CLIC2 was observed at pH 5.0 over the pH range of 5.0–9.0 [19]. However, the selectivity for anions of the channels is very poor and the electrophysiological characterization is not sufficient to call chloride ion channels. Therefore, the question is still being raised as to whether CLICs, including CLIC2, can function as an ion channel. CLICs have homology with GST omega protein and may be involved in regulating ion channels rather than forming ion channels themselves [1,3,36]. The well-known regulatory function of CLIC2 on RyRs support the notion that CLICs modulates channel activities [18]. The three-dimensional structure of human CLIC2 in its water-soluble form has been determined by X-ray crystallography, and two crystal forms have been reported [19]. CLIC2 has an intramolecular disulfide bridge remaining monomeric, whereas an intramolecular disulfide of CLIC1 forms a dimer state. CLIC2 has a highly charged region called foot loop on the C-terminal side. A possibility has been indicated that CLIC2 may interact with other molecules through the highly charged C-terminal region [19]. Although the biological significance of such characteristic molecular structures is still to be elucidated, it may be suggestive of the unique functions of CLIC2, such as the regulation of RyR or binding to MMP14 [4].

### 1.3. Known Functions of CLIC2

This review focuses on CLIC2, which is the least studied of the CLIC family. This paucity of investigation on CLIC2 is partly due to the absence of its gene in mice [10], which has prevented progress in elucidating its function through knockout experiments. CLIC1 and CLIC4 can be inserted into artificial phospholipid membranes to form ion channels with low selectivity under non-physiological acidic conditions [1,46,48]. However, it remains unclear whether they can form ion channels under physiological conditions. Similarly, CLIC2 forms ion channels in artificial membranes with ion conductance similar to that of CLIC4 [46]. However, CLIC2 is scarcely localized in membranous structures including plasma and organellar membranes [4,5], suggesting that the majority of CLIC2 does not form ion channels. Tang et al. [10] have reported that CLICs 1, 4, and 5 can activate NLRP3 inflammasome by their actions as chloride ion channels or as modulators for the ion channels. Since they used murine macrophages, it is not clear whether CLIC2 acts in a similar way as an ion channel.

In humans, *CLIC2* is located at the telomeric region of Xq28, the end of the X chromosome. Human cases with deletions or mutations of this gene, leading to intellectual disabilities predominantly in men, have been reported [48,49]. CLIC2 can bind directly to RyRs while inhibiting its Ca^2+^ channel functions [18,50,51]. One of the mutations of CLIC2 (c.303C>G, p.H101Q) causes the activation rather than inhibition of RyRs, accelerating intracellular Ca^2+^-induced Ca^2+^ release and leading to the abnormal activation of neurons and cardiomyocytes. Cases with this mutation show symptoms of intellectual disability as well as cardiomegaly [48,49]. CLIC2′s only known action at the molecular level is the inhibition of RyRs by directly interacting with them. However, because CLIC2 is widely expressed in various organs and cells (https://www.proteomicsdb.org/proteomicsdb/#human/proteinDetails/O15247/expression), it may have functions other than inhibiting RyRs.

### 1.4. CLIC2 and Malignancy

The findings of a previous report about the relationships between the survival periods of patients with various cancers and expression levels of six CLICs, based on in silico analyses, are summarized in Table 2 [42]. The effects of high or low expression levels of CLICs on the survival rate varies significantly depending on the type of cancer, suggesting that each type of cancer uses different mechanisms for their progression, metastasis, or invasion. In general, the high expression of CLICs 2, 5, and 6 tends to improve the prognosis of cancer, while CLICs 3 and 4 may be detrimental and CLIC 1 appears to have both positive and negative effects. However, only few molecular biological studies [4,5,10,52] have attempted to elucidate how CLICs are involved in cancer progression, making it difficult to clarify the relationship between cancer prognosis and CLICs at the molecular and cellular levels.

Although data from in silico analyses regarding the relationship between CLIC2 and malignancy have recently been published [42,53,54,55], molecular biological studies on the subject are scarce. We recently reported that CLIC2 can prevent malignant cell invasion and distant metastasis [4]. Based on our previous reports and those of others on CLICs, this review discusses the physiological significance of CLIC2 and its role in the pathophysiology of malignant tumors [4,5], particularly focusing on the CLIC2-mediated suppressive regulation of matrix metalloproteinase (MMP) activities and expression. In addition to CLIC2, CLIC4 is also discussed regarding its action on MMPs in Section 7. As shown in Table 1, CLIC4 is a probable detrimental factor for cancer prognosis, and this may be attributable to its stimulating effect on MMP activities, opposite to CLIC2 [29]. Different mechanisms may also underlie the enhancing effects of CLIC4 on malignant tumor growth [52]. Comparing the actions of CLIC2 and CLIC4 may be of help to understand the novel roles of CLICs.

### 1.5. Why CLIC2?

We have studied the mechanism of invasion and metastasis of malignant tumor cells using immunocompetent Wistar rats, where rat glioma cell line C6 cells were transplanted into the subcutaneous tissue in the back [56] or the brain parenchyma [57]. The former is used for distant metastasis to lung analysis, whereas the latter is used for invasion into normal brain tissues surrounding the tumor mass.

The transplantation of C6 cells in the back of Wistar rat neonates results in their death beginning from four weeks after the transplantation due to the hematogenous metastasis of C6 cells from the back tumors to the lungs [56]. Enhanced green fluorescent protein (EGFP)-expressing C6 cells were prepared and transplanted into the back subcutaneous tissue of Wistar rat neonates. Four weeks later, metastatic tumor masses in the lungs were dissected, and the EGFP-expressing C6 cells were isolated by cell sorting (Figure 1). The isolated cells were expanded in vitro before they were re-transplanted into the back of Wistar rat neonates. The gene expression of EGFP-expressing C6 cells from the metastatic lung tumors and primary back tumors was thoroughly examined using RNA-Seq [4]. The gene expression of C6 cells in the metastatic tumors differed markedly from that of C6 cells in the primary tumors, despite the fact that both cells were obtained from the same culture plates. Differentially expressed genes (DEGs) in the metastatic tumors were associated with the cell adhesion, including *periostin* (*Postn*), *cadherin 15* (*Cdh15*), and *dermatopontin* (*Dpt*), as well as angiogenesis-related signaling and regulation of biological quality as the nodes were notably larger in the network. DEGs in the primary tumors were involved in pathways associated with extracellular matrix regulators, such as *chondroitin sulfate N-acetylgalactosaminyltransferase 1* (*Csgalnact1*), and inflammation-related genes such as *oncostatin M receptor* (*Osmr*). (Figure 1). The increase in the expression of inflammation-related genes in the primary tumors may be in accordance with a report demonstrating that CLIC2 improves the prognosis of breast cancer by increasing the number of infiltrated lymphocytes [54]. Therefore, there may be a positive relationship between CLIC2 and anticancer immunity. CLICs 1, 4, and 5 have been linked to NLRP3 inflammasome activation in mouse macrophages, and CLIC2 might exhibit similar proinflammatory activation [10].

Among these diverse gene expression changes, we found that CLIC2 is highly expressed in cells from primary tumors [4]. Because there have been few studies on CLIC2 in relation to malignant tumor progression, the significance and roles of CLIC2 expression in tumor cells were unknown. Therefore, we examined CLIC2 expression in human cancer tissues.

## 2. Expression of CLIC2 in Human Normal and Cancer Tissues

When human primary liver cancer (hepatocellular carcinoma) tissues and normal tissues around cancer tissues were examined for CLIC expression using quantitative RT–PCR, the gene expressions of CLICs 1, 2, 4, and 5 were confirmed, whereas those of CLICs 3 and 6 could not be detected [5]. When the expression levels of the four confirmed CLICs were compared, CLIC1 was found to be the most highly expressed, followed by CLICs 4, 2, and 5, with CLIC5 being expressed at much lower levels. There was no significant difference in the expression of CLICs 1, 4, and 5 between normal and hepatocellular cancer tissues, but CLIC2 was less expressed in cancer tissues than in normal tissue. Even in normal liver tissue, the expression of CLIC2 was significantly decreased in cases of hepatic dysfunction with elevated activities of plasma aspartate transaminase, but there was no significant difference in the expression levels of CLICs 1, 4, and 5. In hepatocellular carcinoma, CLIC2 expression was decreased in cases of advanced stages and/or complicated by liver fibrosis. CLIC2 expression decreased not only in hepatocellular carcinomas but also in colorectal cancer-derived metastatic liver cancers and primary colorectal cancer tissues compared to each surrounding tissue with normal appearance. Western blot yielded comparable results, showing higher CLIC2 protein expression in normal tissues than in cancer tissues. Because the CLIC2 gene is located in the telomere region of the X chromosome [48,49], it was investigated whether sex or age affects CLIC2 expression levels; the results indicated that both sex and age did not affect CLIC2 expression [5].

Immunohistochemical analysis revealed that cancer cells did not express CLIC2. Alternatively, CLIC2 was expressed by blood vessel endothelial cells in the normal tissues surrounding cancer tissues and CD11b-expressing myeloid leukocytes, primarily the monocyte/macrophages and fibroblast-like cells in the cancer stroma. However, most blood vessel endothelial cells in cancer tissues and lymphatic endothelial cells in normal and cancer tissues did not express CLIC2. Such differential expression of CLIC2 at the cellular level may be responsible for the differences in CLIC2 expression levels between normal and cancer tissues.

Despite the notion that CLICs are dimorphic proteins distributed both in soluble and membrane fractions, the localization of CLIC2 in the plasma membrane was hardly found, likely indicating other functions rather than that as an ion channel. CLIC2-expressing blood vessel endothelial cells also expressed claudins 1 and 5, occludin, and ZO-1 (or TJP1), which are tight junction proteins. In contrast, blood vessel endothelial cells in cancer tissues and lymphatic endothelial cells that lacked CLIC2 expression did not express tight junction proteins. These results suggest that CLIC2 is involved in the development and/or maintenance of tight junctions between normal blood vessel endothelial cells. Tight junctions play preventive roles in hematogenous cancer cell metastasis [58,59].

To investigate whether CLIC2 is involved in the suppression of the hematogenous spread of cancer cells, human umbilical vein endothelial cells (HUVECs) that express CLIC2 were seeded to prepare monolayers on the upper chamber of Boyden chambers. SAS cells derived from human tongue squamous cell carcinomas were then seeded on the HUVEC monolayers and the number of cells penetrating the monolayers was counted. SAS cells cannot penetrate normal HUVEC monolayers, but they can penetrate CLIC2-knocked-down monolayers. In that study by Ueno et al. [5], changes in expression and localization of tight junction proteins or basement membranes in HUVECs were not examined when CLIC2 was knocked down. Therefore, the mechanisms underlying the penetration of SAS cells through the CLIC2-knocked-down HUVEC monolayers could not be clarified. CLIC2 can inhibit MMP activities as described below. The absence of CLIC2 may allow MMPs to degrade tight junctions [60] and the basement membrane of blood vessels [61]. Thus, CLIC2 is more highly expressed in normal tissues than in malignant tissues. It may be particularly important for the development and/or maintenance of tight junctions between blood vessel endothelial cells in healthy tissues. Therefore, CLIC2 may play a role in the suppression of the hematogenous spread of malignant cells.

## 3. Suppressive Effects of CLIC2 on Malignant Cell Invasion and Metastasis

If CLIC2 is expressed on tumor cells themselves, what changes in metastasis and invasion of the tumor cells occur? To obtain an answer to this question, C6 glioma cells expressing abundant CLIC2 were established (CC cells). The parent C6 cells did not express CLIC2 protein, but it was highly expressed in the CC cells. CC cells were transplanted into the back of Wistar rat neonates [4]. The proliferative and migrative activities of CC cells were comparable to those of the parent C6 cells. However, CC cells formed smaller primary back tumors and exhibited a much lower incidence of metastatic lung tumor formation. Thus, the rats with CC cell transplants lived much longer than those with parent C6 cell transplants. These findings imply that high CLIC2 expression in primary back tumor cells is correlated with decreased metastasis.

When CC cells are intracerebrally transplanted in Wistar rat neonates to prepare a malignant brain tumor model [57,62], CC cell tumors form smaller tumor masses than parent C6 cell tumors and do not diffusely invade the surrounding normal brain tissues. Consequently, CC cell brain tumors display apparent tumor boundaries, while parent C6 cells form highly invasive large brain tumors with vague boundaries [4]. The results obtained using the two different tumor models show that CLIC2 expression inhibits the invasion and metastasis of malignant cells.

To assess whether the findings from the rat brain tumor model can be applied to humans, CLIC2 expression was investigated in human tissue samples from meningiomas, which are normally benign brain tumors that do not diffusely invade the surrounding brain tissues, and those from glioblastomas, which are the most malignant invasive brain tumors. The most benign grade I meningioma tumors express CLIC2 protein and mRNA at much higher levels than the most malignant grade IV glioblastomas. In meningiomas, CLIC2 was expressed at much higher levels by tumor cells than by blood vessel endothelial cells. It may be worth to note that grade I meningiomas highly express a cell adhesion molecule, E-cadherin [63]. CLIC2 expression was weaker in the rather benign grade I glioma cells than in grade I meningioma cells but still higher than that in grade IV glioblastoma cells. In meningioma cases, the association between progression-free survival (PFS) and CLIC1, CLIC2, and CLIC4 expression levels was examined, and PFS was found to be significantly prolonged only when CLIC2 was highly expressed. Although the number of patients studied was small, PFS was significantly prolonged in glioblastoma cases when CLIC2 was highly expressed. The findings from databases on survival and CLIC2 expression in gastric and lung cancer are compatible with these clinical data on brain benign and malignant tumors [42].

Based on the CLIC2 expression level, 10 human glioma stem cell lines were divided into two groups (high and low CLIC2 expression), and RNA-Seq was used to determine the gene expression of each group. The cell lines with low CLIC2 expression demonstrated increased expression of genes related to glycolysis, hypoxia, and the mesenchymal pathway [4], while those with high CLIC2 expression demonstrated an increased expression of genes related to the adhesion-related signaling and proneural pathways. These results suggest that CLIC2 is upregulated in association with benign tumor traits.

## 4. Vascular Permeability and CLIC2

Increased vascular permeability is associated with the hematogenous metastasis of cancer [64,65], and the tight junction proteins are closely related to the inhibition of this vascular permeability [5,66]. Studies on human cancer suggest that CLIC2 inhibits hematogenous metastasis by strengthening adhesions between vascular endothelial cells [5], and similar results have been obtained in a rat model of distant metastasis [4]. When the blue dye Evans blue was injected through the tail vein into rats with parent-C6-cell back and lung metastatic tumors, both tumors turned blue. In contrast, the back tumors of rats transplanted with CC cells showed no blue staining. This indicates that the expression of CLIC2 in tumor cells as well as in endothelial cells may prevent the increase in vascular permeability.

To investigate how CLIC2 expression in tumor cells correlates with vascular permeability, the expression of various cell adhesion molecules was examined. E-cadherin protein levels increased in CC cell primary tumors, but there was no significant difference in tight junction proteins. Because there was no significant difference in the expression of mRNA encoding E-cadherin, the difference in protein levels may be due to differences in translation or the degree of protein degradation. The analysis of proteins in tumor masses from human meningioma grade I and grade II cases showed increased VE-cadherin protein in the more benign grade I meningiomas. Furthermore, in the same samples, MMP2 activation was suppressed in grade I tumor masses with abundant CLIC2 protein expression. The activated MMP2 was more abundant in grade II cells than in grade I cells. Grade II cells exhibited higher gelatinolytic activity than grade I cells.

Surprisingly, the Western blot analysis of culture supernatants revealed that grade I meningioma cells secreted a significant amount of CLIC2 (Figure 2). A small amount of CLIC4 was detected in the culture supernatant, but no CLIC1 was detected. To investigate the possibility that CLIC2 was secreted exclusively by exosomes, the exosomal fraction was subjected to Western blotting; however, the results revealed that CLIC2 was undetectable, suggesting that CLIC2 is released from the secretory granules to the extracellular milieu through exocytosis. CLIC2 is intracellularly colocalized with vesicle-associated membrane protein 7 (VAMP7), a secretory granule marker, confirming that CLIC2 is a secretable protein. CLIC2 was also localized in the Golgi apparatus, and MMP2 and MMP14 (MT1-MMP) were colocalized in VAMP7-positive secretory granules with CLIC2.

It has been reported that CLIC2 is diffusely distributed in the cytoplasm and nucleus of the human embryonic kidney 293 cell line at a steady state [67]. When these cells are stimulated with G-protein coupled receptor (GPCR) agonists such as lysophosphatidic acid or acetylcholine, CLIC2 moves to the vicinity of plasma membrane around cell adhesion structures that may be associated with the actin cytoskeleton. CLIC4 shows similar changes in localization. This localization change to the vicinity of the plasma membrane might be correlated with the extracellular secretion of the CLICs.

## 5. Roles of Secreted CLIC2; Relationship with MMPs

CLIC2 was not detectable in the culture supernatant of parent C6 cells, but MMP2 was abundant. CLIC2 was found in high concentrations in the culture supernatant of CC cells, whereas MMP2 was found in low concentrations. MMP14, which activates MMP2 by partially degrading proMMP2, was not expressed differentially in C6 and CC cells. Active MMP14 is a plasma membrane-associated protein, but in CC cells, MMP14 localization to the plasma membrane was reduced, while secretion to the extracellular space was significantly increased, implying that CLIC2 may inhibit MMP14 plasma membrane localization. It was found that CLIC2 binds to MMP14 by immunoprecipitation, albeit weakly. Recombinant CLIC2 protein prepared by a cell-free wheat germ protein synthesis system inhibited MMP14 activity in a concentration-dependent manner, and the inhibitory effects were comparable to those of the same concentration of N-Isobutyl-N-[4-methoxyphenylsulfonyl] glycyl hydroxamic acid, a broad-spectrum and water-soluble MMP inhibitor. The inhibitory effect of CLIC2 on MMP14 activity was stronger than that of the tissue inhibitor of metalloproteinase 2 (TIMP2), an endogenous inhibitory protein for MMPs. The mechanism of MMP14 inhibition by CLIC2 and TIMP2 may be similar because no synergistic effect was observed when TIMP2 and CLIC2 were mixed and added to the MMP14 activity assay system.

MMP14 is responsible for malignant cell invasion and metastasis by activating MMP activities. Then, can the CLIC2 protein prevent invasion and metastasis by inhibiting MMP14 activities? CLIC2 protein was added to the culture medium during an invasion assay with U251 human glioblastoma cell line using the Boyden chamber with Matrigel-coated cell culture inserts. CLIC2 significantly reduced the invasion of U251 cells, despite the cells demonstrating strong invasive activity by degrading the Matrigel in the absence of CLIC2. Similar preventive effects on invasion were observed when parent C6 cells or another human glioblastoma cell line SFC-2 was used. The recombinant CLIC4 protein did not show any suppressive effects on the invasion of parent C6 cells. CLIC4 is highly expressed by many tumor cells, but it did not show inhibitory effects on MMP14.

Primary cultured human meningioma cells, which originally expressed CLIC2 abundantly, showed little invasive activity, but the knockdown of CLIC2 resulted in marked invasion in the Boyden chamber assay. This knockdown experiment suggests that the suppressive effects of CLIC2 on MMP activities are exerted both intracellularly and extracellularly (Figure 2), but it is unclear which one plays a more central role. This issue can be resolved through experiments such as suppressing the action of CLIC2 outside the cell by adding neutralizing antibodies.

However, the observation that recombinant CLIC2 can inhibit malignant cell invasion by suppressing MMP activity helps to explain why normal tissues and benign tumors do not undergo metastatic invasion. CLIC2 or related endogenous mechanisms may be used to create new therapeutic strategies against the invasion and metastasis of malignant tumors.

## 6. Intercellular Adhesive Structures and CLIC2

There are three possible mechanisms by which CLIC2 suppresses distant metastasis of malignant cells. The first possibility is that CLIC2 causes the stabilization of intercellular adhesion between normal blood vessel endothelial cells, thereby preventing the invasion of malignant cells into circulation. In cancer tissues, scarce CLIC2 expression may lead to unstable intercellular adhesion, enabling the hematogenous spread of malignant cells. The second possibility is that tumor cells express CLIC2 that binds to MMP14 intracellularly while inhibiting the enzyme activity. The third possibility is that tumor cells secrete CLIC2 that inhibits MMP activities in the extracellular milieu, resulting in the maintenance of intercellular adhesion and extracellular matrix leading to the prevention of invasion and metastasis of tumor cells.

MMPs are responsible for the destruction of blood vessel tight junctions, which leads to the disruption of the blood–brain barrier (BBB) [68]. For example, MMP activities are elevated in ischemic brain lesions, and an MMP9 inhibitor reduces stroke volume [69], leading to the prevention of BBB disruption [60]. Normal blood vessels in the brain express CLIC2, which may suppress MMP activities in homeostatic conditions, leading to the maintenance of tight junctions in BBB. Various claudins, on the other hand, activate MMP2 through MMP14 [70]. This finding suggests that tight junctions activate MMPs, which cause tight junction disruption, implying that MMPs must be homeostatically suppressed by some mechanism. CLIC2 is a likely candidate for suppressing MMP activities to maintain barrier functions of normal blood vessels. Rat models of stroke or traumatic brain injury accompanied by increased vascular permeability and elevated MMP activities may be required to investigate changes in CLIC2 expression levels. It is worthwhile to investigate whether the administration of CLIC2 into circulation can prevent BBB breakdown in rodent ischemic or traumatic injury models.

CLICs 1 and 4 also stimulate barrier functions of endothelial cells through the activation of small guanosine triphosphatase (GTPase) Rac1 or RhoA in response to sphingosine 1-phosphate [71]. Although this is a distinct mechanism from the CLIC2-mediated mechanism through the inhibition of MMPs, the resultant enhanced barrier function of blood vessels leads to the suppressed invasion and metastasis of malignant cells. However, CLICs 1 and 4 are also involved in angiogenesis [28,72] and the enhancement of sprouting of blood vessel endothelial cells [28,73]. Therefore, CLICs 1 and 4 may be involved in both the suppression and progression of malignancy regarding blood vessels.

## 7. Modulation of MMP Activities by CLIC4

As described above, CLIC2 inhibits tumor cell invasion and metastasis by suppressing MMP14 activity. On the other hand, CLIC4 may stimulate the progression of malignant tumors [42,52]. CLIC4, a more widely expressed CLIC than CLIC2, has an opposite effect on MMP14 [29]. CLIC4 is reportedly colocalized with MMP14 in late endosomes of a human normal retinal pigment epithelial-derived cell line, ARPE19 cells. CLIC4 may play a role in the maintenance of MMP14 activity. MMP14 is localized in lipid rafts, which are detergent-resistant cholesterol-rich membrane microdomains on plasma membrane. However, when CLIC4 is knocked down in ARPE19 cells, MMP14 localization in lipid rafts is prevented. An immunoprecipitation assay revealed that CLIC4 can bind to MMP14. Furthermore, CLIC4 stimulates the extracellular secretion of MMP2. When ARPE19 cells are cultured on gelatin, the cells show gelatinolytic activities, which are abolished if CLIC4 is knocked down. Thus, CLIC4 may be necessary for the activities of MMPs. The report by Hsu et al. [29] is the first one to show the capability of CLIC to modulate MMP activities. CLIC2 binds to MMP14 and regulates MMP2 activity, indicating the similarity between CLIC2 and CLIC4 in terms of the modulation of MMP activities. It is necessary to further investigate whether CLICs are generally responsible for the regulation of MMP activities. The observation that CLIC2 inhibits MMPs, which is the opposite of what CLIC4 does, is particularly intriguing. As shown in Table 1, generally, CLIC4 exerts detrimental effects on cancer progression, while CLIC2 shows ameliorative effects. The distinct effects of CLICs on cancer progression may be associated with their different effects on MMPs. Although mice lack CLIC2, CLIC5 and CLIC6 may be the CLIC genes that can substitute CLIC2 in mice. It is necessary to investigate the effect of CLIC5 and CLIC6 on MMP activities in the future to reveal whether CLIC5 or CLIC6 can substitute the suppressive CLIC2 functions on MMPs. C6 glioma cells strongly express CLIC4, and it will also be necessary to examine whether the forced expression of CLIC2 and the knockdown of CLIC4 have a synergistic suppressive effect on invasion and metastatic potential.

## 8. Conclusions

Analyses of databases have shown that the CLICs affect the prognosis of malignancy. CLIC 4 tends to worsen the prognosis, whereas CLICs 2 and 5 improve it. However, the underlying molecular and cellular biological mechanisms remain largely unknown. Except for its inhibitory effects on RyR, CLIC2 has received little attention as only a few molecular and cellular biological studies have been conducted on CLIC2. This review focused on the effect of CLIC2 on malignancy, although information on the subject is still limited.

CLIC2 can be considered a favorable CLIC for malignant tumor prognosis. The mechanism by which CLIC2 improves the tumor prognosis involves maintaining tight junctions and cadherin-mediated adherence junctions of blood vessels, which inhibit hematogenous distant metastasis. Even in brain tumors, which do not cause distant metastasis, CLIC2 suppresses the invasiveness of the tumor cells into the surrounding brain tissues. CLIC2 inhibits MMP14 activities through mutual binding, preventing MMP2 activation and degradation of ECM proteins. Consequently, adhesions between blood vessel endothelial cells are maintained, and the hematogenous spread of malignant cells is prevented. MMP14 inhibition may suppress ECM degradation, which is required for malignant cell invasion and metastasis. In contrast to CLIC2, CLIC4 binds to MMP14 and activates it to promote cell migration and invasion [29]. Only CLIC2 and CLIC4 have been reported to be involved in the regulation of MMP activity to date, but other CLICs should be re-examined from this perspective. For example, CLIC1 may be able to regulate MMP14 activity, as CLIC1 possess the same amino acid sequence necessary for binding to MMP14 as CLIC4 based on analysis by Hsu et al. [29]. In particular, because mice lack CLIC2, it is assumed that some of the other CLICs may be substituted for the role of CLIC2. Thus, the binding of other CLICs to MMP14 as well as their other properties should be investigated.

Although CLICs have been identified as ion channels, their characteristic subcellular localization, such as within secretory granules or just below the plasma membrane, and their extracellular secretion make it unlikely that CLICs function as ion channels, and the name CLIC may thus need to be reconsidered.

## 9. Prospects

It is crucial to investigate new CLIC2-based therapies for malignant tumors because the increased expression of CLIC2 can prevent the metastatic invasiveness and distant metastasis of malignant tumors. This could be implemented using the CLIC2 protein itself or by gene transfer. If some factors that increase CLIC2 expression are discovered, they could be used as pharmacological interventions to prevent malignant cell invasion and metastasis by increasing CLIC2 expression. The factors that regulate CLIC2 expression levels in normal and tumor tissues must be identified. Despite having the same origin, C6 cells isolated from primary back tumors expressed CLIC2 at much higher levels than C6 cells in culture dishes. On the other hand, CLIC2 expression levels in C6 cells isolated from metastatic lung tumors were almost identical to those in the cultured C6 cells. It is critical to determine whether the significant changes in CLIC2 expression are the cause or result of C6 cell invasion and metastasis. The following questions should be addressed. Do the cells not metastasize as a result of increased CLIC2 expression, or do they metastasize as a result of unchanged CLIC2 expression? Does the increase in CLIC2 expression or unchanged expression result from remaining in the primary tumor or from metastasizing? The results of transplanting CC cells into the neonatal rat back suggest that increased CLIC2 expression prevents metastasis. Currently, there are no available data providing insight into extracellular signals, such as cytokines, affecting CLIC2 expression. It is still elusive whether different culture conditions for C6 cells can affect the expression of CLIC2.

However, if CLIC2 expression can be increased in malignant tumor tissues and/or surrounding tissues, there is a strong possibility that cell–cell junctions, particularly those between vascular endothelial cells, will be strengthened and hematogenous metastasis will be suppressed. Additionally, it is anticipated to prevent invasion into neighboring tissues even in malignancies such as brain tumors that do not spread to other locations. The forced expression of CLIC2 by transfection, administration of the CLIC2 protein itself, or molecules that induce CLIC2 expression are expected to become novel therapeutic interventions to prevent invasion and metastasis of malignant cells. CLIC2 expression is suppressed by the presence of fetal bovine serum or TGFβ1 in primary cultured fibroblasts from human mammary glands [74]. When stimulated with ligands for GPCRs, such as tachykinin NK2 receptors, muscarinic M3, and lysophosphatidic acid receptors, CLIC2 translocates from the cytoplasm to the vicinity of the plasma membrane and cell adhesion structures [67]. Such reports indicate that some pharmacological interventions can increase or decrease CLIC2 expression or alter its subcellular localization. This further indicates that CLIC2 could be a new target molecule in cancer therapy. With respect to MMP14 activity, CLIC2 and CLIC4 have opposite effects, and CLIC2 may act in an inhibitory and CLIC4 in a promotive manner with respect to the metastasis and invasion of malignant tumors. Therefore, it is possible that the therapeutic effect on malignant tumors will be further enhanced if not only the effect of CLIC2 is enhanced but also that of CLIC4 is inhibited at the same time.

In cerebral infarction, the BBB is disrupted due to increased MMP activity, exacerbating the prognosis [60]. MMP inhibition therapy, where CLIC2 or related substances can be administered intravenously, may be easily applicable in acute diseases, such as cerebral infarction. MMPs are also involved in a wide variety of other diseases, including musculoskeletal diseases such as rheumatoid arthritis, periodontitis, multiple sclerosis, fibrotic lung disease, arteriosclerosis, and glomerulonephritis [75,76]. CLIC2 could be used to regulate MMP activity in the pathophysiology of these various diseases.

Because CLIC is conserved well in a wide variety of metazoans [1,2,3], it could be involved in the regulation of various cellular activities. CLIC2 and CLIC4 should be further investigated because they may play a role in the regulation of universal life phenomena. Because MMPs are also involved in a remarkably broad range of biological phenomena [77], it should be further explored whether CLIC2, CLIC4, and possibly other CLICs are also involved in the control of common biological phenomena.

## Figures and Tables

**Figure 1 cancers-14-04890-f001:**
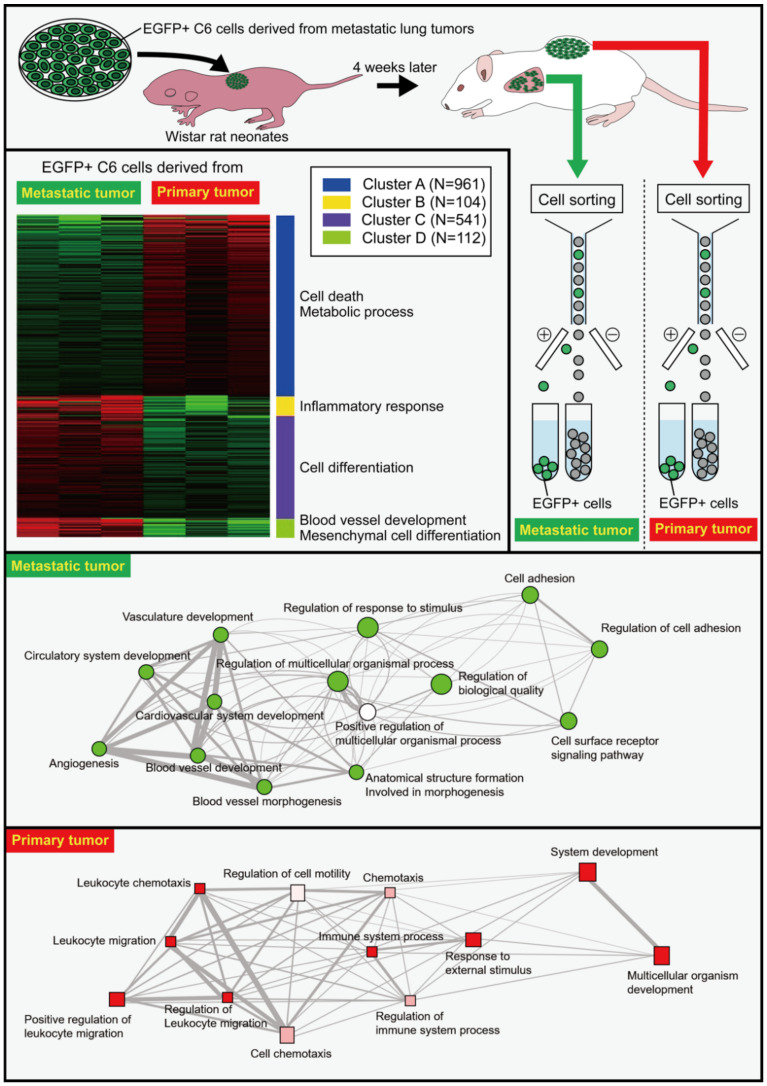
Differential gene expression in cells derived from metastatic and primary tumors as revealed by RNA-Seq. EGFP-expressing C6 glioma cells were transplanted in the back of Wistar rat neonates. Four weeks later, the metastatic tumor masses were dissected before the isolation of EGFP^+^ C6 cells from the tumor masses using a fluorescence-activating cell sorter (FACS). The isolated cells were cultured to expand the cell number, and the expanded cells were re-transplanted into the back of Wistar rat neonates. EGFP^+^ cells were isolated from lung metastatic tumors and back primary tumors four weeks later. Isolated cells were subjected to RNA-Seq analysis that revealed marked differences between the cells from the metastatic and the primary tumors. Gene ontology analyses showed the differential pathways of genes between metastatic and primary tumors in biological processes. For more detailed information, see the literature by Ozaki et al. [4].

**Figure 2 cancers-14-04890-f002:**
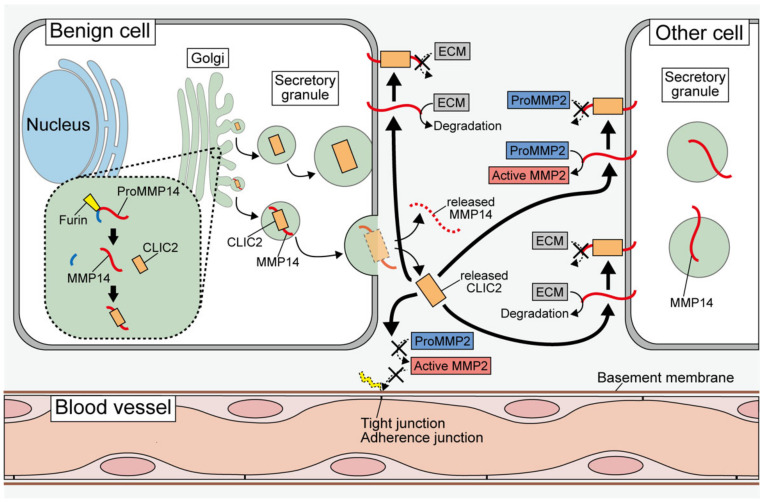
Probable mechanisms underlying the CLIC2-mediated suppression of tumor cell invasion and metastasis. CLIC2 is localized in the Golgi apparatus and secretory granules in benign tumor cells such as meningiomas. In the secretory granules, CLIC2 binds to MMP14 and inhibits the localization of MMP14 in the plasma membrane. Thus, proMMP2 is not activated by membrane-bound active MMP14. Furthermore, extracellular matrix (ECM) proteins are not degraded because of the inactive MMP2 and MMP14, and tight junctions and adherence junctions are not destroyed. Released CLIC2 binds to MMP14 on the surface of other cells while preventing the activation of MMP2 and ECM degradation. Thus, there may be two different mechanisms: intracellular and extracellular ones. For more detailed information, see papers [4,5].

**Table 1 cancers-14-04890-t001:** CLICs: their distribution, ion channel activities, and biological functions.

CLICs	Distribution	Ion Channel	Biological Function	References
CLIC1	Various organs	Poorly selective anion channels	Participates in inflammatory processes Activation of MYC signaling Enzymatic activity Redox regulation	[11,12,13,14,15,16,17]
CLIC2	Blood vessels. heart, liver	Anion channels	Modulation of ryanodine receptor Inhibition of MMP14 activity	[4,5,13,18,19]
CLIC3	Muscles, heart, lung, kidney	Component of anion channel, regulator of channel	Endosomal trafficking Promote invasive behavior	[20,21,22,23]
CLIC4	Various organs	Poorly selective ion channels	Enhance tumor invasiveness Enhance TGF-β signaling Induce apoptosis Involved in angiogenesis Stimulation of MMP14 activity	[24,25,26,27,28,29]
CLIC5	Kidney, heart, lung, colon	Poorly selective ion channels	Actin cytoskeleton-dependent membrane remodeling	[9,25,30]
CLIC6	Soluble and membrane fractions	Unknown	Interact with dopamine receptors	[31,32,33]

**Table 2 cancers-14-04890-t002:** Correlation between CLIC expression levels and cancer mortality. Effects of high CLIC expression levels on the survival of patients with cancer are simply summarized based on our own interpretation of the data reported by Gururaja Rao et al. [42]. n.s., not significant.

Cancer	Breast	Ovarian	Lung	Gastric	Liver	Pancreatic
**CLIC1**	detrimental	detrimental	n.s.	ameliorative	detrimental	detrimental
**CLIC2**	ameliorative	detrimental	ameliorative	ameliorative	ameliorative	n.s.
**CLIC3**	detrimental	detrimental	n.s.	detrimental	n.s.	detrimental
**CLIC4**	n.s.	detrimental	detrimental	detrimental	n.s.	detrimental
**CLIC5**	ameliorative	detrimental	ameliorative	ameliorative	n.s.	n.s.
**CLIC6**	ameliorative	ameliorative	ameliorative	ameliorative	n.s.	n.s.

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
