# Peer review of "Chloride Intracellular Channel Proteins (CLICs) and Malignant Tumor Progression: A Focus on the Preventive Role of CLIC2 in Invasion and Metastasis"

_cancers, 2022, doi:10.3390/cancers14194890_

Round 1

Reviewer 1 Report

The review on CLIC2 by Ozaki et al is timely and well written. CLICs are known to play an important role in cancer onset and progression. There are several studies linking CLICs with a diagnosis of cancer. So far CLIC4 is the most extensively studied and published CLIC protein. The review highlights the importance of CLIC2. There are several gaps that should be filled.

1. Sentence 194 is misleading as the authors mention that since CLIC2 is soluble and is present in the nucleus it does not function as an ion channel. CLIC1 is also present in soluble form and nucleus but does form a functional channel. There are reports on the activity of CLIC2 in bilayers. The authors should correct this statement to highlight that CLIC2 is also a dimorphic protein (soluble and ion channel form).

2. There should be a section on the structure of CLIC2 (soluble) and a presence of a trans membrane domain (for membrane form). This will provide a clear picture of CLIC2 for readers.

3. CLIC2 is a regulatory protein for RyRs and several of the functions mentioned in the review involves Ca2+. The authors should have provided a possible mechanism mediated by CLIC2 by regulating Ca2+ in these processes.

4. Expression of CLIC2 in different types of cancers is not uniform. This should be discussed in detail.

5. CLICs are anion channels and in the review importance of ions in mentioned cellular processes is not discussed. This is important to highlight the importance of anion channels for physiology.

6. Section 7 is abrupt for the introduction of CLIC4. There should be a clear rationale to include CLIC4 in the regulation of MMP activity. What about other CLICs?

7. Sentence 401, why CLIC1 cannot be a substitute for CLIC2 as CLIC1 is present in the nucleus and is also present in the soluble form?

Author Response

Response to <Reviewer 1>

The review on CLIC2 by Ozaki et al is timely and well written. CLICs are known to play an important role in cancer onset and progression. There are several studies linking CLICs with a diagnosis of cancer. So far CLIC4 is the most extensively studied and published CLIC protein.

The review highlights the importance of CLIC2. There are several gaps that should be filled.

[Thank you for your supporting comments. We also greatly appreciate for the following critical and constructive comments. Our responses are in parentheses.]

  1. Sentence 194 is misleading as the authors mention that since CLIC2 is soluble and is present in the nucleus it does not function as an ion channel. CLIC1 is also present in soluble form and nucleus but does form a functional channel. There are reports on the activity of CLIC2 in bilayers. The authors should correct this statement to highlight that CLIC2 is also a dimorphic protein (soluble and ion channel form).

[In response to this comment and the next one, we have revised the sentence and added the statement that CLIC2 is a dimorphic protein in the new section (1.2. Known structures of CLIC2). Although this is a valuable convincing comment indicating that CLICs are the dimorphic protein, we could not clearly find localization of CLIC2 in the plasma membrane as well as intracellular membranes by our immunohistochemical study. Furthermore, our own subcellular fractionation study could not identify significant CLIC2 localization in membrane fraction, although this result has not been published. Based on our own results and the paucity of published convincing evidence for the membranous localization of CLIC2 in vivo, we mainly addressed CLIC2 in the soluble fractions in this review.]

  1. There should be a section on the structure of CLIC2 (soluble) and a presence of a transmembrane domain (for membrane form). This will provide a clear picture of CLIC2 for readers.

[In response to this request, we have newly added “a section 1.2. Known structures of CLIC2” to describe the characteristic structures of CLIC2.]

  1. CLIC2 is a regulatory protein for RyRs and several of the functions mentioned in the review involves Ca2+. The authors should have provided a possible mechanism mediated by CLIC2 by regulating Ca2+ in these processes.

[Although CLIC2 is involved in Ca2+ homeostasis through regulation of RyRs, we do not have any evidence in either of our own experiments or literature for involvement of CLIC2 in Ca2+ regulation in relation to pathophysiological processes of cancers. Therefore, we think it difficult to describe the relationship between CLIC2 and calcium beyond what is stated in the introduction.]

  1. Expression of CLIC2 in different types of cancers is not uniform. This should be discussed in detail.

[There are no published reports describing differences in CLIC2 expression levels in different kinds of carcinomas in a way that allows direct comparison, making it difficult to describe more than what is currently described. We have evaluated CLIC2 expression in liver and colon cancers as well as meningioma and glioma including glioblastoma. However, because different methods (different kit and agents) were employed, we could not directly compare the CLIC2 expression levels between liver/colon cancers and the brain tumors.]

  1. CLICs are anion channels and in the review importance of ions in mentioned cellular processes is not discussed. This is important to highlight the importance of anion channels for physiology.

[We would like to discuss the relationship between CLICs and pathophysiology of cancers. However, there are none of the papers showing the direct involvement of CLICs in the cancer pathophysiology through activities as anion channels. We think that the physiological role of CLICs in general was mentioned in the introduction. General functions of CLICs have been repeatedly documented in many review articles, and therefore, we focused novel functions of CLIC2 in pathophysiological processes in cancers.]

  1. Section 7 is abrupt for the introduction of CLIC4. There should be a clear rationale to include CLIC4 in the regulation of MMP activity. What about other CLICs?

[CLIC4 and CLIC2 are the only CLICs that have been reported of their positive and negative effects on MMPs. As Reviewer says, section 7 may be abrupt. Therefore, we added some description of CLIC4 regarding the involvement of regulation of MMP activities. We wish to say that some interventions to increase CLIC2 expression and decrease CLIC4 expression at the same time may be a promising novel therapy for malignancy.]

  1. Sentence 401, why CLIC1 cannot be a substitute

[Thank you for this critical and interesting comment. However, we can't add anything at present, as so far the only CLICs that have been noted to be related to MMP are CLIC4 and CLIC2.]

Reviewer 2 Report

Summary

This is a very clearly written and highly informative review that describes novel findings regarding the diverse elusive roles of chloride intracellular channel proteins (CLICs). Recent studies have revealed that CLICs which are not plasma membrane delimited should receive a different designation since their functions are far more complex than merely serving as chloride conduits. Instead in silico analyses indicate that CLICs appear to be prognostic markers of cancer malignancy. CLIC4 upregulation appears to have an adverse effect on cancer prognosis whereas CLIC2 and CLIC5 have an opposite effect. Despite this realization the underlying molecular and cellular biological mechanisms are for the most part unknown. This review focuses for the most part on the favorable effects of CLIC2 in inhibiting tumor malignancy.  CLIC2 is secreted extracellularly and binds to matrix metalloproteinase, which inhibits MMP14 activity and inhibits MMP2 activation and degradation of other tight junctional ECM proteins. In turn these effects improve tumor prognosis through maintaining tight junction integrity, which suppresses inhibit hematogenous distant metastasis. These effects of CCL2 warrant further assessment as a potential target to treat tumor prognosis. CLIC4 also binds to MMP14 but activates it to promote tumor cell migration and invasion. Therefore, CLIC 4 tends to worsen the prognosis, whereas CLICs 2 and 5 improve it. Only CLIC2 and CLIC4 have been reported to be involved in the regulation of MMP activity to date, but other non-plasma membrane delimited CLICs warrant further study as potential targets to improve cancer therapeutics.

Author Response

Response to <Reviewer 2>

There are no specific comments made by Reviewer 3.

[We sincerely appreciate your supporting high rating.]

Reviewer 3 Report

The authors of this manuscript have reviewed the potential role of CLIC2 channels to prevent the metastasis of cancer cells. Some improvements need to be made in order to make this review more comprehensive.

Major comments:

  1. Some sentences are written in vague language. For example, in line#17 "CLIC2s are relatively new channel proteins". What does this mean? Have they been recently discovered? Even if that is what the authors meant, this is not true (many studies have reported the role and function of CLIC2 genes/proteins in the late 1990s).
  2. Section 1.1 -> The authors do not make it clear if and which CLICs are considered to be ion channels. Importantly, it should be mentioned if they are ligand- (like calcium-activated chloride channels - see PMID: 28121009) or voltage-gated ion channels. To be consistent with the tile of the manuscript, I suggest adding a table and summarizing the role of each class of CLICs in this introductory section.
  3. Figure 1 can be shortened to focus only on the main findings (and not the methodology). Figure 2 seems to be redundant and is just a duplicate of the authors' original research article, hence needs to be removed.
  4. Section 7 emphasizes the modulatory role of CLIC4 on MMPs. Throughout the manuscripts, authors refer to different classes of CLICs instead of focusing on CLIC2 itself. This trend could be confusing for authors. I suggest revising the text to focus only on CLIC2 and removing the sections/paragraphs that are out of CLIC2 scope.

Minor Edits:

  1. Figure1 -> Refenerncee is missing in the figure caption.
  2. Figure3 -> Refenerncee is missing in the figure caption.
  3. The acknowledgment section needs to be edited.

Author Response

Response to <Reviewer 3>

The authors of this manuscript have reviewed the potential role of CLIC2 channels to prevent the metastasis of cancer cells. Some improvements need to be made in order to make this review more comprehensive.

[We greatly appreciate for the following critical and constructive comments. Our responses are in parentheses.]

Major comments:

  1. Some sentences are written in vague language. For example, in line#17 "CLIC2s are relatively new channel proteins". What does this mean? Have they been recently discovered? Even if that is what the authors meant, this is not true (many studies have reported the role and function of CLIC2 genes/proteins in the late 1990s).

[Thank you for the kind comment. We have removed the indicated sentence and rewrote the first two sentences in Abstract. We also revised unclear or inaccurate wording throughout the manuscript.]

  1. Section 1.1 -> The authors do not make it clear if and which CLICs are considered to be ion channels. Importantly, it should be mentioned if they are ligand- (like calcium-activated chloride channels - see PMID: 28121009) or voltage-gated ion channels. To be consistent with the tile of the manuscript, I suggest adding a table and summarizing the role of each class of CLICs in this introductory section.

[In our knowledge, it has not been clarified whether gating of CLICs is regulated by some ligands or changes in voltage or some other mechanisms. There are still only a few knowledge especially in case of CLIC2 whether they are really working as anion channel. As described in the manuscript, the majority of intracellular CLIC2 is present within secretory granules and they are extracellularly secreted. Therefore, we think that the central role of CLIC2 may not be forming ion channels.

Despite a valuable suggestion to add new Table listing the roles of CLICs, many critical and prevalent reviews have already carried such kinds of Tables and/or Figures (see section 1. 2.; For review on ion channel functions of CLICs). Therefore, we introduced such reviews for readers, as an alternative to ad new Tables. This is also because we would like to focus on CLIC2 as we said in response to the comment 4.]

  1. Figure 1 can be shortened to focus only on the main findings (and not the methodology). Figure 2 seems to be redundant and is just a duplicate of the authors' original research article, hence needs to be removed.

[Thank you for the critical comments. Despite the suggestion, we think it would be better to keep Figure 1 with its description on the methodology. This is because the experiments are rather complicated, and the methodology Figure would be of good help for readers to enhance understanding. Regarding Figure 2, in response to the suggestion, we removed it.]

  1. Section 7 emphasizes the modulatory role of CLIC4 on MMPs. Throughout the manuscripts, authors refer to different classes of CLICs instead of focusing on CLIC2 itself. This trend could be confusing for authors. I suggest revising the text to focus only on CLIC2 and removing the sections/paragraphs that are out of CLIC2 scope.

[CLIC2 inhibits metastasis and invasion of cancer by suppressing MMPs, while CLIC4 aggravates cancer by increasing MMP activity. We believe that by describing these contrasting actions, we can highlight the function of CLIC2. Moreover, we wish to say a possibility that some interventions to increase CLIC2 expression and decrease CLIC4 expression at the same time may be a promising novel therapy for malignancy. We hope you can understand our intentions. Furthermore, there is still limited information at the molecular and cellular levels on CLICs in relation to the pathophysiology of cancer, which makes it difficult to write a review article without such a comparison. In response to this comment, we have deleted probably confusing and redundant sentences from introduction. ]

Minor Edits:

  1. Figure1 -> Refenerncee is missing in the figure caption.

[Thank you for pointing this out. We added reference.]

  1. Figure3 -> Refenerncee is missing in the figure caption.

[Thank you for pointing this out. We added reference.]

  1. The acknowledgment section needs to be edited.

[We are quite sorry for our mistake. We revised the section appropriately.]

Round 2

Reviewer 3 Report

The authors have partially addressed my comments on the original version of the manuscript. The current version can still be shortened by removing redundant figures. Adding summary tables would greatly help with the ease of readability of the text. The manuscript, as it is, merely summarizes the current body of knowledge and lacks a critical perspective needed to investigate the anti-/pro-metastatic role of CLICs in cancers.

Author Response

The authors have partially addressed my comments on the original version of the manuscript. The current version can still be shortened by removing redundant figures. Adding summary tables would greatly help with the ease of readability of the text. The manuscript, as it is, merely summarizes the current body of knowledge and lacks a critical perspective needed to investigate the anti-/pro-metastatic role of CLICs in cancers.

[We greatly appreciate for the critical and constructive comments. Our responses are in parentheses.]

[In response to the request to add summary Table, we newly add summary Table as Table 1. Table 1 in the previous manuscript was changed to Table 2 in the new manuscript.]

[In response to the comment, we added a proposal in the last section (line 518~) for a new malignancy treatment that takes advantage of the anti-/pro-metastatic properties of CLIC2 and CLIC4.]

Round 3

Reviewer 3 Report

The authors have addressed the concerns I raised. Thanks

Author Response

Response to <Reviewer 3>

Please carefully check the spelling. There are not a large number of mistakes, but the first paragraph in the abstract already contains two (articles too many), and I have seen more throughout the text. Please consider if it is feasible recapitulating the data in summary tables. (mainly careful spell and grammar check of the sections newly added).

[We greatly appreciate for the comments. We have asked professionals to correct the grammatical and typographical errors. We have attached certificate provided by the professionals. We think we have recapitulated the data in Table 2. ]
